# Cannabinoid CB1 Receptor Deletion from Catecholaminergic Neurons Protects from Diet-Induced Obesity

**DOI:** 10.3390/ijms232012635

**Published:** 2022-10-20

**Authors:** Raj Kamal Srivastava, Inigo Ruiz de Azua, Andrea Conrad, Martin Purrio, Beat Lutz

**Affiliations:** 1Institute of Physiological Chemistry, University Medical Center of the Johannes Gutenberg University of Mainz, 55128 Mainz, Germany; 2Department of Zoology, Indira Gandhi National Tribal University, Amarkantak, Anuppur 484887, India; 3Leibniz Institute for Resilience Research (LIR), 55122 Mainz, Germany

**Keywords:** CB1 receptor, catecholaminergic neurons, neuropeptide Y, norepinephrine, obesity

## Abstract

High-calorie diets and chronic stress are major contributors to the development of obesity and metabolic disorders. These two risk factors regulate the activity of the sympathetic nervous system (SNS). The present study showed a key role of the cannabinoid type 1 receptor (CB1) in dopamine β-hydroxylase (*dbh*)-expressing cells in the regulation of SNS activity. In a diet-induced obesity model, *CB1* deletion from these cells protected mice from diet-induced weight gain by increasing sympathetic drive, resulting in reduced adipogenesis in white adipose tissue and enhanced thermogenesis in brown adipose tissue. The deletion of *CB1* from catecholaminergic neurons increased the plasma norepinephrine levels, norepinephrine turnover, and sympathetic activity in the visceral fat, which coincided with lowered neuropeptide Y (NPY) levels in the visceral fat of the mutant mice compared with the controls. Furthermore, the mutant mice showed decreased plasma corticosterone levels. Our study provided new insight into the mechanisms underlying the roles of the endocannabinoid system in regulating energy balance, where the CB1 deletion in dbh-positive cells protected from diet-induced weight gain via multiple mechanisms, such as increased SNS activity, reduced NPY activity, and decreased basal hypothalamic-pituitary-adrenal (HPA) axis activity.

## 1. Introduction

Obesity and obesity-related disorders, a major health concern of current societies, are associated with high-calorie diet intake and chronic stress. Indeed, daily hypercaloric diets create a conducive environment for the development of metabolic syndrome and obesity, characterized by an increase in body weight, visceral fat deposition, and insulin resistance, with subsequent increased risk for cardiovascular diseases. Chronic stress exposure is often associated with weight gain and increased levels of the stress hormone glucocorticoid [1]. However, interindividual differences are commonly observed, but the mechanisms underlying the different metabolic outcomes have remained elusive.

The sympathetic nervous system (SNS) has been implicated in the regulation of body weight and energy homeostasis, as well as in stress responses (reviewed in [2,3]). The catecholaminergic neurons are mainly located in the locus coeruleus (LC) in the brain stem. Catecholaminergic neurons located outside of the LC are present in the nucleus of the solitary tract (NTS) and the ventrolateral medulla (VLM). In addition, sympathetic post-ganglionic neurons and chromaffin cells of the adrenal medulla also synthesize and secrete catecholamines (reviewed in [4]).

The activated SNS has a protective role in the development of obesity through the release of norepinephrine (NE) from sympathetic nerve terminals that innervate adipose tissues [5,6,7,8,9]. In brown adipose tissue (BAT), NE release enhances energy expenditure and thermogenesis through the activation of β3-adrenergic receptors [10], by activating lipases and enhancing the expression of uncoupling protein-1 (Ucp1). In white adipose tissue (WAT), NE activates β3-adrenergic receptors and promotes lipolysis, which is required as a source of energy for thermogenesis in BAT and other tissues, thereby counteracting the adiposity [11].

Stress responses involve the activation of the sympatho-adrenal system, with concomitant activation of the hypothalamic–pituitary–adrenal (HPA) axis. The sympatho-adrenal system secretes stress-related hormones such as NE, epinephrine, and neuropeptide Y (NPY). NPY, which is secreted not only from the adrenal medulla, but also from the central and peripheral catecholaminergic neurons, has been implicated in stress and metabolic responses [12]. Activation of the HPA axis leads to increased glucocorticoid secretion from the adrenal cortex. Notably, anatomical evidence has shown that the adrenal cortex is innervated by sympathetic fibers and inputs from adrenal medullary ganglion cells [13]. However, contrary to the catabolic effects of the stress hormones NE and epinephrine, increased glucocorticoid secretion has been linked to the development of idiopathic obesity and metabolic syndrome, which shares many comorbidities with Cushing’s syndrome, characterized by hypercortisolism [14,15]. Indeed, chronically elevated plasma glucocorticoid levels promote energy storage in the adipose tissue, particularly in visceral adipose tissue [16].

The endocannabinoid (eCB) system is crucially involved in the regulation of energy homeostasis through the activation of CB1, both in the central and peripheral tissues [17,18]. Importantly, obesity has been extensively associated with an elevated eCB system tone [19,20]. Consequently, genetic deletion of CB1 in mice protects from diet-induced obesity [21,22,23,24,25]. Several studies have shown that the pharmacological and genetic CB1 blockade can modulate energy homeostasis by increasing energy expenditure through enhanced thermogenesis [23,25,26,27,28]. In addition, the eCB system also controls stress responses through the modulation of the HPA axis activity [29,30,31], among other mechanisms. Accordingly, it has been established that glucocorticoids can stimulate eCB synthesis, which is essential for the glucocorticoid-mediated negative feedback of the HPA axis activity [32,33].

Therefore, we investigated whether the loss of CB1 specifically in catecholaminergic (dbh-expressing) cells affects diet-induced obesity.

## 2. Results

### 2.1. Characterization of Mice Lacking the CB1 Receptor in dbh-Expressing Cells

In order to delete the *CB1* gene from dopamine-β-hydroxylase (*dbh*)-expressing cells, a mouse line expressing Cre recombinase under the regulatory sequences of the *dbh* gene (dbh-Cre mouse) [34] was crossed with the *CB1* floxed mouse line [35]. The dbh-Cre mouse line expresses Cre recombinase in the sympathetic ganglia, adrenal medulla, and catecholaminergic neurons in the LC, NTS, and VLM [34]. Mice lacking *CB1* in *dbh*-expressing cells are called dbh-CB1-KO (KO or mutant), and the control floxed littermates lacking the Cre recombinase expression are termed dbh-CB1-WT (WT or control). This mutant mouse line has previously been investigated regarding memory consolidation after stress [36] and bone development [37].

The deletion of *CB1* in *dbh*-expressing cells was confirmed by the qPCR analysis of mRNA isolated from sympathetic superior cervical ganglia (SCG), revealing a more than 95% reduction of *CB1* mRNA levels compared with the control littermates (Figure 1A). Furthermore, in the adrenal gland, a 50% reduction in *CB1* mRNA levels was observed [36]. However, qPCR when performed on isolated adrenal medulla showed a larger reduction (~75%) in *CB1* mRNA levels [36]. Brain punches from LC showed a significant reduction in *CB1* mRNA levels (Figure 1B). The remaining *CB1* mRNA is most likely a contribution of non-catecholaminergic cells. The deletion of *CB1* in LC was further demonstrated by double fluorescent in situ hybridization labeling *dbh* and *CB1* mRNA, confirming *CB1* deletion selectively in *dbh*-positive cells (Figure 1C). Thus, these results demonstrate the selective deletion of the *CB1* gene in *dbh*-expressing cells.

### 2.2. dbh-CB1-KO Mice Show Reduced Weight, Adiposity, and Increased BAT Thermogenesis on a High-Fat Diet

HFD was reported to activate the SNS as well as the eCB system [26,38,39]. Therefore, we challenged both dbh-CB1-KO and -WT mice with HFD treatment. We found that dbh-CB1-KO mice gained less weight than dbh-CB1-WT in HFD conditions, despite the similar daily food intake (Figure 2A,B). Moreover, the dbh-CB1-KO mice on HFD showed an improved metabolic plasma profile, including reduced levels of plasma leptin (Figure 2C), insulin (Figure 2D), triglycerides, and low-density lipoprotein (LDL)/cholesterol (Figure 2E,F). Hepatic triglyceride accumulation was also decreased in KO compared with WT mice (Figure 2G).

Consistent with the lean phenotype and improved metabolic plasma profile, dbh-CB1-KO mice on HFD displayed less visceral fat and a reduced adipocyte size (Figure 2H–J). The gene expression analysis of markers in the visceral WAT showed reduced mRNA levels of *leptin (Lep), fatty acid synthase (Fas), and CCAAT/enhancer-binding protein β (Cebpb)* mRNA in dbh-CB1-KO mice compared with the control mice (Figure 2K). Gene expression analysis of thermogenic markers in BAT revealed increased mRNA levels of *uncoupling protein (Ucp1), transcription factor A (Tfam) and Lipe* (encoding hormone-sensitive lipase, Hsl) in dbh-CB1-KO mice compared to dbh-CB1-WT (Figure 2L).

### 2.3. dbh-CB1-KO Mice on a High-Fat Diet Show Increased Energy Expenditure and SNS Tone

Then, we analyzed the energy expenditure in these mice through indirect calorimetry. In agreement with the analysis of the thermogenic markers in BAT, the dbh-CB1-KO mice on HFD had increased oxygen consumption (Figure 3A), carbon dioxide production (Figure 3B), heat production (Figure 3C), and energy expenditure both during light and dark phases (Figure 3E,F). However, no differences in total ambulation (Figure 3D) were observed. ANCOVA analysis of the energy expenditure using body weight as a covariate showed significantly increased heat dissipation in dbh-CB1-KO mice compared with the control littermates (Figure 3G). In conclusion, HFD-fed dbh-CB1-KO mice showed reduced weight gain, predominantly caused by reduced adipogenesis in the visceral fat and a higher energy expenditure, most likely through enhancement of the β3-adrenergic signaling in the adipose tissue mediated by an increased SNS tone.

In BAT, we analyzed the NE turnover, as a measure of the SNS tone, and *β3 adrenergic receptor* mRNA levels in these mice on HFD (Figure 3H). We observed that the dbh-CB1-KO displayed a significantly increased NE turnover (i.e., SNS tone) and enhanced *β3 adrenergic receptor* mRNA levels compared with the WT controls, which might be responsible for the leaner phenotype.

### 2.4. Lack of CB1 in dbh-Positive Neurons Enhances SNS Activity While Decreases NPY Activity in the Visceral Fat under Obesogenic Conditions

Next, we further detailed the NE and NPY activity in dbh-CB1-KO mice in obesogenic conditions. There is evidence that sympathetic neurons synthesize and co-secrete neuropeptides, such as NPY [40,41,42,43]. However, NPY exerts the opposite effect on adipose tissue expansion compared with NE. While NPY promotes the proliferation of adipocyte precursors via NPY type 1 receptor (NPY1r) [44], NE increases lipolysis and thermogenesis. To this end, we analyzed the changes in NE and NPY activity in HFD-fed mice (Figure 4). Dbh-CB1-KO mice showed reduced plasma NPY levels, and reduced *Npy* and *Npy1r* mRNA levels in visceral fat compared with the control littermates, suggesting decreased NPY signaling in the visceral fat (Figure 4A,B).

Further analysis of the visceral fat showed increased NE turnover as well as enhanced *β1, β2, and β3 adrenergic receptor (β1AR, β2AR, and β3AR)* mRNA levels, indicating enhanced NE signaling in the visceral fat (Figure 4C,D), thus protecting dbh-CB1-KO mice against HFD-induced obesity. A well-accepted readout of NE signaling is the phosphorylation of hormone-sensitive lipase [45,46,47]. Immunoblot analysis of visceral fat showed a significant increase in the phosphorylation of Hsl (p-Hsl/Hsl) in HFD-fed dbh-CB1-KO mice (Figure 4E,F). Additionally, dbh-CB1-KO showed higher plasma NE levels than the WT mice (Figure 4G). Recent studies have suggested that the activation of β3AR might lead to the upregulation of the eCB tone, thus limiting the increase in energy expenditure [48,49]. We found that *CB1* and *Ucp1* mRNA levels in the visceral fat of dbh-CB1-KO mice were not significantly different compared with the WT control littermates (Figure 4H,J). However, the *CB1* mRNA expression was significantly increased in the BAT of dbh-CB1-KO mice compared with the WT control littermates (Figure 4I).

Taken together, CB1 deficiency in *dbh*-positive cells upregulated NE signaling and decreased NPY signaling in the visceral WAT upon HFD treatment compared with the WT mice, thus protecting dbh-CB1-KO mice against HFD-induced obesity.

### 2.5. HFD-Fed dbh-CB1-KO Mice Showed Reduced Proliferation and Angiogenesis in the Visceral Fat and Decreased HPA Axis Activity

Adipose tissue expansion has been associated with increased proliferation and angiogenesis. The immunostaining for the proliferation marker, Ki67, indicated a tendency to have reduced proliferation in the visceral fat of dbh-CB1-KO mice (Figure 5A,B). Angiogenesis markers, such as angiopoietin 1 (Angpt1) and thrombospondin 1 (Thbs1), are associated with obesity [50]. The low *Angpt1* expression and high expression of the endogenous angiogenesis inhibitor *Thbs1* in dbh-CB1-KO suggested reduced angiogenesis (Figure 5C). Thus, decreased angiogenesis in dbh-CB1-KO mice on HFD might contribute to the lower expansion of adipose tissue.

Besides playing a protective role against the development of obesity, the sympatho-adrenal system modulates the HPA axis activity [13]. Therefore, we evaluated the contribution of corticosterone (CORT) in the observed metabolic phenotype of dbh-CB1-KO mice. We observed that HFD-fed dbh-CB1-KO mice showed lower plasma CORT levels than WT control mice (Figure 5D). Accumulating evidence has demonstrated that elevated CORT levels are linked to obesity and metabolic syndrome [16,51], hence, at this point, we cannot exclude that low CORT levels in mutant mice also contribute to protection against diet-induced obesity.

## 3. Discussion

The present study provides new insights into the role of *CB1* in *dbh*-expressing cells in diet-induced obesity. In this obesity model, the deletion of *CB1* from these cells increased the SNS activity in visceral WAT and BAT, thereby protecting against HFD-induced obesity.

Accumulating evidence has shown that the activation of SNS improves metabolic health by promoting energy expenditure and lipolysis in adipose tissues [2,11,52,53]. In humans, thermogenically active BAT is detectable in lean subjects [54], indicating that modulating energy expenditure and thermogenesis might be an attractive way to tackle obesity. The sympathetic neurons innervate various fat depots [55]. Thus, NE released from sympathetic terminals can bind to β3-adrenergic receptors expressed on adipocytes and promote lipolysis [7,11] in WAT and thermogenesis in BAT [9].

Several studies have shown that genetic and pharmacological CB1 blockade increases sympathetic drive in adipose tissues, protecting against diet-induced obesity [23,25,56,57,58]. Moreover, the activation of presynaptic CB1 in sympathetic nerve endings inhibits NE release [59]. Based on these observations, it has been assumed for a long time that peripheral CB1 expression in SNS terminals regulates sympathetic tone in peripheral tissues, but this hypothesis has not yet been directly addressed. In the present study, the selective deletion of CB1 from sympathetic neurons increased sympathetic tone in mutant mice in obesogenic conditions. In these mice, we observed that adipogenesis was reduced in the visceral fat, while thermogenic markers were increased in BAT. Accordingly, mutant mice showed a higher energy expenditure, but no differences in food intake and locomotor activity. Furthermore, dbh-CB1-KO mice on HFD showed a significant increase in SNS activity (NE turnover) and *β-adrenergic receptor* expression in BAT. Thus, the deletion of *CB1* from catecholaminergic neurons rendered mutant mice resistant to diet-induced obesity due to increased SNS activity and energy expenditure, without affecting the locomotor activity and food intake. In contrast, β3 adrenergic receptor activation or cold exposure has been previously shown to increase the eCB tone in BAT and EWAT [48,49]. However, in the present study, *CB1* mRNA levels were only significantly different, i.e., increased, in BAT compared with the WT control mice.

Hypercaloric diets lead to the development of obesity and metabolic syndrome, both characterized by expanded visceral fat [60,61,62]. In the present study, the lack of *CB1* in catecholaminergic neurons suppressed visceral fat expansion. In particular, the reduced visceral fat in the mutant mice was due to decreased hypertrophy, adipogenesis, and angiogenesis in the visceral fat. The observed adipose tissue remodeling in the mutant mice on HFD was associated with decreased circulating NPY levels and decreased local *Npy* and *Npy1r* gene expressions in the visceral fat. Our findings suggest that the decreased NPY activity in visceral WAT contributes to decreased proliferation, adipogenesis, and angiogenesis, as observed in dbh-CB1-KO mice. Several findings support this notion, as, for example, subcutaneous delivery of NPY stimulates body weight, fat mass, and the vascularization of visceral fat in mice [63]. In addition, in in vitro models, NPY stimulates the proliferation of mouse 3T3-L1 preadipocyte cells, rat preadipocytes, and endothelial cells [44,63]. Nevertheless, it is unclear whether the reduction of visceral fat is mediated by decreased circulating NPY levels or by decreased local *Npy* expression in adipose tissues. Supporting the local synthesis of NPY in adipose tissue, *Npy* expression has also been reported in stromal vascular fraction (SVF) from the same fat depot [64]. Importantly, *Npy* expression has been found to be elevated in the SVF and adipose tissue macrophages of obese mice [44,65]. Recently, a study confirmed that peripheral NPY1r antagonism or selective ablation of *NPY1r* from adipocytes increases thermogenesis and protects from diet-induced obesity [66]. Thus, the decreased NPY signaling in adipose tissues reduces adipogenesis and increases energy expenditure.

Furthermore, dbh-CB1-KO mice on HFD showed a significant increase in SNS outflow (NE turnover), *β-adrenergic receptor* expression, and sympathetic activity in the visceral fat. Conversely, the reduced β-adrenergic signaling might promote severe obesity, as observed in the mice lacking *β-adrenergic receptors* [5]. Thereby, the reduced NPY signaling and concomitant increased NE signaling in visceral fat decreased the abdominal fat deposition in mutant mice during diet-induced obesity.

Finally, we also observed a decrease in circulating CORT levels in dbh-CB1-KO mice on HFD. As chronically elevated plasma glucocorticoid levels have been strongly associated with fat mass expansion [16,51], the reduced plasma CORT levels may also contribute to the decreased visceral fat in mutant mice. Anatomical studies indicate that the *Crh*-positive neurons in the paraventricular nuclei of the hypothalamus, the primary driver of the HPA axis, receive inputs from catecholaminergic neurons in the brainstem [64,67,68]. Additionally, several investigations have demonstrated that glucocorticoids mobilize eCBs for the suppression of the HPA axis activity via central CB1 activation [32,33]. Furthermore, adrenal cortex secretion is regulated by sympathetic innervation and adrenal medullar innervation under stress and non-stress conditions [13]. Therefore, the lack of CB1 in catecholaminergic neurons may also modulate the glucocorticoid release from the adrenal cortex via different mechanisms, hence contributing to the glucocorticoid-mediated metabolic changes in mutant mice.

In conclusion, our study unravels a novel mechanism of the role of the eCB system in obesity and metabolic syndrome involving the *CB1* expression in *dbh*-positive cells. Hereby, the deletion of *CB1* in catecholaminergic neurons protected from diet-induced weight gain by increasing SNS activity and reducing NPY activity in adipose tissue, concomitant with changes in the circulating CORT levels, suggesting a potential role of peripheral CB1 in obesity and obesity-related disorders. Our findings are in agreement with other studies that suggest a protective role of the peripherally restricted CB1 inverse agonist in obesity by reducing appetite, body weight, and leptin resistance [27,28].

## 4. Material and Methods

### 4.1. Animals

We generated mice lacking the CB1 in *dbh*-expressing cells by crossing *CB1* floxed female mice [69] with *CB1* floxed male mice expressing Cre recombinase under the regulatory sequences of the *dbh* gene [34]. All of the experimental animals were genotyped for the presence of homozygous *CB1* floxed allele and heterozygous *dbh-Cre* allele by PCR using the following primers: for *CB1* floxed, G50: forward 5′-GCT GTC TCT GGT CCT CTT AAA, G51: reverse 5′-GGT GTC ACC TCT GAA AAC AGA; for Cre recombinase: forward 5′-GCG TCA GAG ATT TGT TGG AGG AC, reverse 5′-CAC AGC ATT GGA GTC AGA AGG G. The *Cre*-negative littermates (containing the *CB1* floxed/floxed alleles) were used as the wild-type controls. Importantly, the use of only male *Cre*-positive mice in the breeding prevented the possible germ line deletion of *CB1*. The mice were kept single-housed at a controlled temperature (22 ± 1 °C) and humidity (45 ± 10) with a 12 h light/dark cycle.

### 4.2. Diet Experiments

Male mice, 7–8-weeks-old, singly housed, had free access to water and were fed ad libitum with a high-fat diet (HFD) for 3 months. In HFD, the caloric contribution was 60% fat, 17% protein, and 23% carbohydrate (Altromin, Lage, Germany; C1090-60). Body weight and food intake were recorded twice a week. At the end of the diet experiments, the mice were fasted and sacrificed by decapitation under isoflurane, and the trunk blood was collected in EDTA-coated tubes with a protease inhibitor, cooled on the ice, and immediately centrifuged at 1000× *g* for 5 min. The plasma was collected and kept frozen at −80 °C until it was analyzed for insulin, leptin, and lipids. The visceral fat was removed and weighed. Other tissues such as BAT, liver, brain, and sympathetic cervical ganglia were also removed and snap-frozen at −80 °C until analyzed. We used separate cohorts of mice for monitoring energy expenditure, NE turnover, and tissue collection. All of the experiments were carried out following the European council directive and were approved by the Ethical committee on animal care and use of Rhineland-Palatinate, Germany.

### 4.3. Plasma Hormone and Lipid Analyses

Plasma insulin and leptin were measured in overnight-fasted mice on HFD. Insulin was measured using ultrasensitive immunoassay kits (Alpco diagnostics, Salem, NH, USA; 80-INSMSU-E01). Leptin was measured using immunoassay kits from Millipore, Burlington, MA, USA (EZML-82K). The plasma total cholesterol and LDL cholesterol were measured using commercially available kits (BioVision, Waltham, MA, USA; K631-100). Plasma and hepatic triglycerides were measured by a triglyceride quantification kit (Bio Vision; K622-100). The plasma CORT was measured in mice (5–6-months-old) on HFD. Blood sampling was done within 30 s of mouse handling by rapid submandibular bleeding. The plasma CORT levels were measured following the instructions described in the kit (IBL International, Hamburg, Germany; RE52211).

The plasma NPY levels were measured in mice on HFD. Blood was quickly collected from the submandibular vein in EDTA vials with protease inhibitors within 30 s of mice handling. As platelets are a rich source of NPY, blood was centrifuged at 2000× *g* for 15 min at 4 °C to separate the plasma and platelets. The plasma was kept frozen at −80 °C until assayed. NPY in platelet-free plasma was measured by an enzyme-linked immunoassay kit (Millipore, Burlington, MA, USA; EZRMNPY-27K).

### 4.4. Indirect Calorimetry Measurements

Energy expenditure was measured by indirect calorimetry in the metabolic cages (TSE systems GmbH, Bad Homburg, Germany) six weeks after HFD treatment to monitor the metabolic activity over a 48-h period. Mice were acclimatized for 24 h in the cages before the start of the actual measurements. After the acclimatization, heat production, O_2_ consumption, CO_2_ production, locomotor activity, and food and water intake were measured every 15 min for the next 48 h. Energy expenditure was analyzed through an analysis of the covariance (ANCOVA) using body weight as a covariate in order to exclude the potential effect of body weight differences in the measurements [70].

### 4.5. Norepinephrine Turnover

NE turnover was determined in the adipose tissues of mice on HFD, through the administration of α-methyl-p-tyrosine (AMPT; Sigma, St. Louis, MO, USA; M8131), a competitive inhibitor of tyrosine hydroxylase, the rate-limiting enzyme in NE synthesis, as described earlier [71]. The mice from each genotype were divided into two groups, the first group was sacrificed, and adipose tissues were collected to measure basal level (time = 0). The other group was injected intraperitoneally with AMPT (0.25 mg/g BW), and the second dose of AMPT (0.125 mg/g BW) was injected 2 h after the first injection to maintain the inhibition of tyrosine hydroxylase. The mice were killed 4 h after the first AMPT injection and the adipose tissues were collected, snap-frozen in liquid nitrogen, and stored at −80 °C until assayed.

The NE tissue content was measured using an enzyme-linked immunoassay kit (LDN diagnostic, Nordhorn, Germany; BA E-5200). Briefly, ~100–150 mg of tissues were homogenized in a buffer containing 0.01 M HCl, 1 mM EDTA, and 4 mM sodium disulfite. After centrifugation for 10 min (10,000× *g*, 4 °C), NE was extracted from the supernatant following the instructions supplied with the ELISA kit. Finally, the NE turnover was calculated using the following formula: NE turnover = k[NE]_0_, where k = (lg[NE]_0_ − lg[NE]_4_)/(0.434 × 4), [NE]_0_ = basal NE concentration, and [NE]_4_ = final NE concentration [72].

### 4.6. Histology and Immunohistochemistry of Adipose Tissue

The tissues were fixed in paraformaldehyde (4% in PBS), dehydrated in graded ethanol, cleared, and embedded in paraffin. Then, 10 µm thick sections were obtained and mounted on the slides. The slides were deparaffinized and stained with hematoxylin (Sigma, St. Louis, MO, USA; MHS16) and eosin (Sigma, St. Louis, MO, USA; 2853), as per the standard procedure. Sections were observed under a Leica DMRA microscope (Leica Microsystems, Wetzlar, Germany). For immunohistochemistry, the slides were deparaffinized and rehydrated with graded alcohol. After quenching the endogenous peroxidase, the tissue sections were blocked with blocking serum (VECTASTAIN Elite ABC Kit; Vector Laboratories, New York, NY, USA; PK-6102). Tissue sections were incubated with an anti-Ki-67 primary antibody (1:1000, Ki-67 mouse mAb; Cell Signaling, Danvers, MA, USA; 9449) overnight at 4 °C. The tissue sections were washed and detected using a secondary antibody from VECTASTAIN Elite ABC Kit (Vector Laboratories, New York, NY, USA; PK-6102). The images were captured with a Leica DMRA microscope (Leica Microsystems, Wetzlar, Germany) and the staining was quantified by ImageJ software (ImageJ version 1.45, NIH, Bethesda, MD, USA).

### 4.7. Real-Time qPCR

Tissue samples and brain punches were homogenized in 1 mL TRIzol (Invitrogen, Waltham, MA, USA; 15596-018). The total RNA was extracted and purified by RNeasy kit (Qiagen, Hilden, Germany; 74106) following the instructions supplied with the kit. cDNA was synthesized from 500 ng of total RNA using a high-capacity cDNA reverse transcription kit (Applied Biosystems, Waltham, MA, USA; 4390778). qPCR was performed using the Taqman gene expression master mix (Applied Biosystems, Waltham, MA, USA; 4369510) and the Taqman probes described in Table 1. All of the reactions were run in duplicate, and the expression levels of mRNA were calculated by the comparative Ct method normalized to the TATA-binding protein (*Tbp*). The mRNA expression levels in the neural tissues were normalized to glucuronidase beta (*Gusb*).

### 4.8. Immunoblot

The adipose tissue samples were homogenised in a RIPA buffer (50 mM Tris 8.0, 0.5% sodium deoxycholate, 0.1% SDS, 150 mM NaCl, 1% Triton X-100) containing protease and phosphatase inhibitors (Thermo Scientific, Waltham, MA, USA; 78445). Samples were cleared from lipids by using methanol and chloroform followed by TCA precipitation from the aqueous phase. For immunoblot, 2 μg of total protein was separated by SDS-PAGE and transferred to a nitrocellulose membrane. To detect p-Hsl, the membranes were blocked (5% milk in TBS-T buffer) and incubated overnight with rabbit anti-p-Hsl antibody (1:1000; Cell Signaling, Danvers, MA, USA; 4126) in the same blocking buffer. To detect the total Hsl nitrocellulose, the membranes were stripped with stripping buffer (2% SDS, 100 mM TCEP-HCl, 62.5 mM Tris 6.8) and the stripped membranes were again blocked (5% milk in TBS-T) and incubated with primary antibody against the total Hsl (1:1000; Cell Signaling, Danvers, MA, USA; 4107). The immunoreactivity was detected by peroxidase-conjugated anti-rabbit antibody and an enhanced chemiluminescence Western blot reagent kit (ECL, GE Healthcare, Amersham, UK). Glyceraldehyde 3-phosphate dehydrogenase (*Gapdh*) was used as loading control.

### 4.9. In Situ Hybridization

The mice were killed by decapitation under isoflurane and their brains were isolated, quickly frozen on dry ice, and stored at −70 °C until being used for sectioning. Sections of 20 µm thickness were cut on a cryostat. The DIG-labelled riboprobes for CB1 and dbh were prepared as described earlier [73]. The signal amplification was achieved by TSA plus system cyanine3/fluorescein (Perkin Elmer, NEL 744001KT, NEL 741001KT). The images were captured using a Leica DMRA microscope (Leica Microsystems, Wetzlar, Germany).

### 4.10. Statistical Analysis

All of the results were represented as mean ± SE and were statistically analyzed using Prism software (GraphPad version 8, San Diego, CA, USA). Statistical analyses were performed by Student’s *t*-test, or one-way or two-way analysis of variance (ANOVA), followed by Bonferroni post-hoc test, as indicated in the figure legends. ANCOVA analyses of the energy expenditure were performed using body weight as a covariate using SPSS 24 (IBM Corporation, New York, NY, USA).

## Figures and Tables

**Figure 1 ijms-23-12635-f001:**
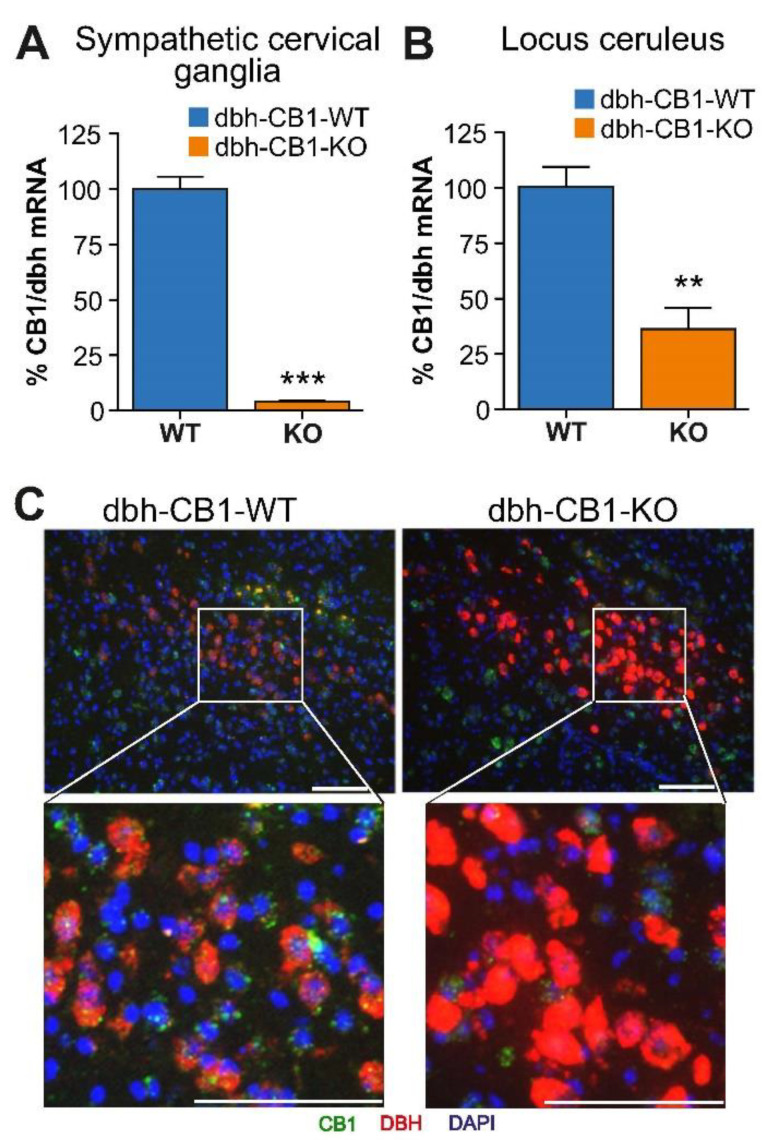
Characterization of mice lacking the CB1 receptor in *dbh* expressing cells. (**A**) qPCR analysis of *CB1/**dbh* ratio mRNA levels in the superior cervical ganglia (*n* = 5). (**B**) qPCR analysis of *CB1**/dbh* ratio mRNA levels in brain punches (0.5 mm diameter) from the locus ceruleus brain region with surrounding tissues. Data are mean ± SE of *n* = 3–4 samples. (**C**) Double fluorescent in situ hybridization in locus ceruleus. Green: *CB1* receptor, Red: DBH, Blue: DAPI. Scale bar = 50 μm. For (**A**,**B**), Student *t*-test; ** *p* ≤ 0.01, *** *p* ≤ 0.001.

**Figure 2 ijms-23-12635-f002:**
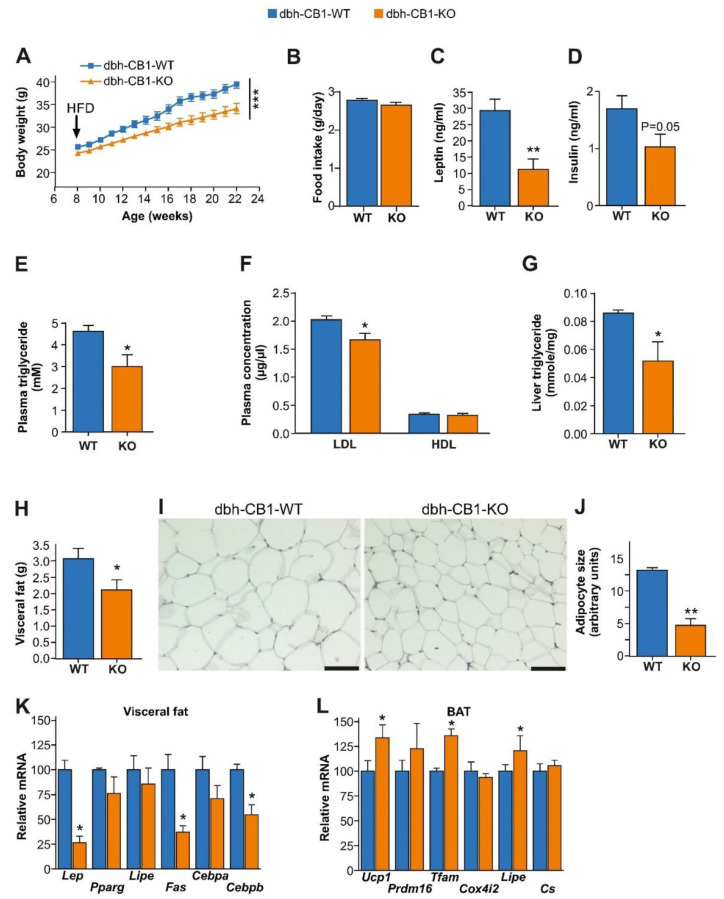
dbh-CB1-KO mice show a reduced body weight gain, increased thermogenic markers in BAT, decreased fat mass, and adipogenesis markers in visceral fat. (**A**) dbh-CB1-KO mice are resistant to HFD-induced obesity compared with WT (*n* = 15 each). *** *p* ≤ 0.001 by repeated measures two-way ANOVA. (**B**) dbh-CB1-KO mice show no differences in food intake (*n* = 15; Student *t*-test). (**C**) dbh-CB1-KO show decreased plasma leptin level (*n* = 8). ** *p* ≤ 0.01 by Student *t*-test. (**D**) Fasting plasma insulin levels in mice on HFD. (**E**) Lower plasma triglyceride levels in dbh-CB1-KO mice on HFD. (**F**) Lower LDL cholesterol levels in dbh-CB1-KO mice on HFD but no differences in HDL cholesterol levels compared with WT mice (*n* = 7–8). * *p* ≤ 0.05 by student *t*-test. (**G**) HFD-fed dbh-CB1-KO mice show significantly low liver triglyceride tissue levels (*n* = 4–5). * *p* ≤ 0.05 by student *t*-test. Data are expressed as mean ± SE. (**H**) dbh-CB1-KO show decreased visceral fat (*n* = 8–9). * *p* ≤ 0.05 by Student *t*-test. (**I**) Hematoxylin and eosin staining on histological sections from visceral fat tissue. Scale bar = 100 μm. (**J**) Quantitative analysis of adipocyte size (*n* = 3). ** *p* ≤ 0.01 by Student *t*-test. (**K**) dbh-CB1-KO show decreased *Lep, Fas, and Cebpb* mRNA levels in visceral fat whereas no significant differences are observed in the other markers of adipogenesis. (**L**) dbh-CB1-KO show increased *Ucp1, Tfam, and Lipe* in BAT. For (**K**,**L**) (*n* = 4), * *p* ≤ 0.05 by Student *t*-test. Data are expressed as mean ± SE.

**Figure 3 ijms-23-12635-f003:**
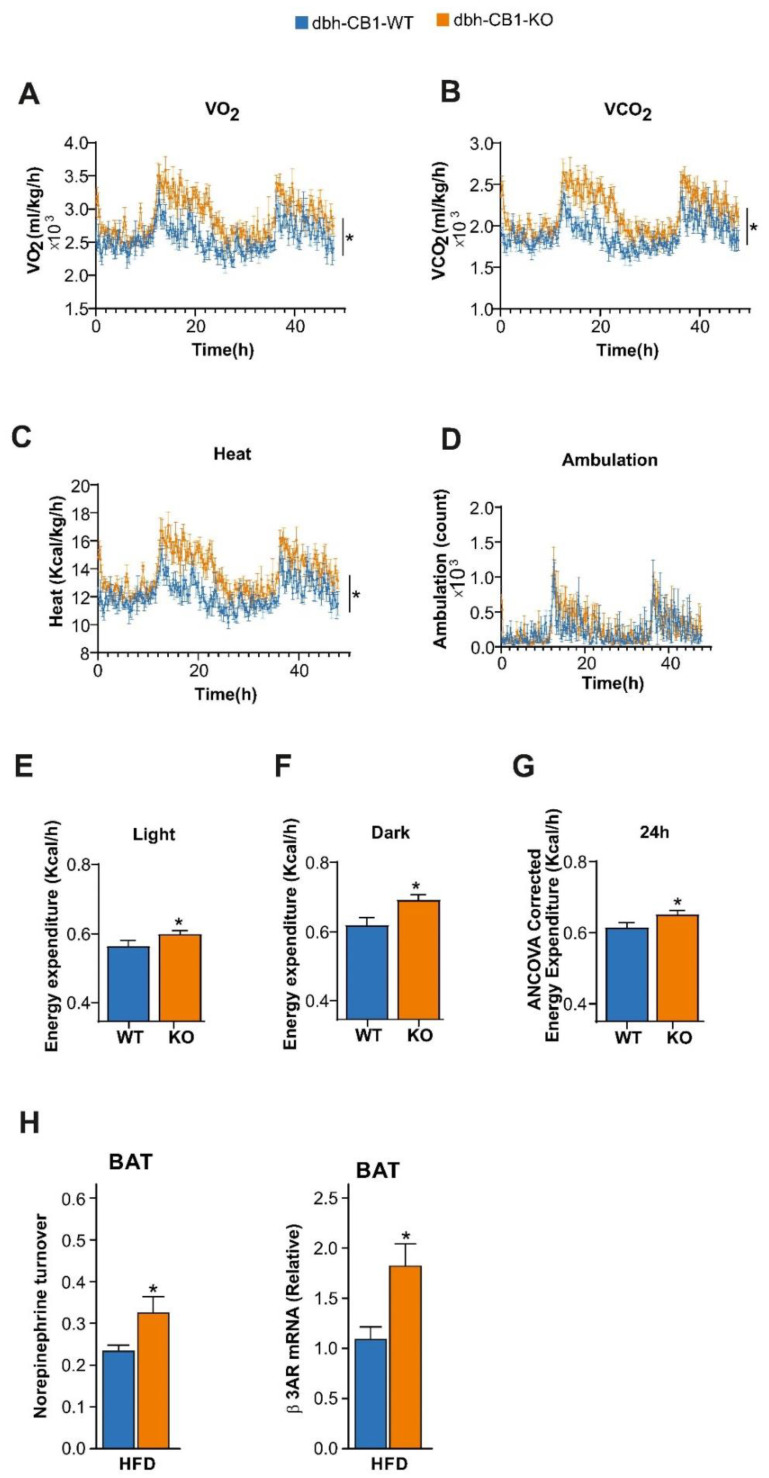
dbh-CB1-KO mice show increased energy expenditure on HFD. (**A**) dbh-CB1-KO mice show increased oxygen consumption and (**B**) increased CO_2_ release. (**C**) Heat production was increased although no significant statistical difference was observed in (**D**) ambulation in KO mice as compared with WT control (*p* ≤ 0.05 by two-way ANOVA; *n* = 5–8). dbh-CB1-KO mice showed high energy expenditure both in (**E**) light and (**F**) dark phase, and (**G**) ANCOVA analysis of the energy expenditure was calculated using body weight as the covariate. (**H**) Norepinephrine turnover (NETO) in the BAT of dbh-CB1-KO mice is increased after HFD treatment as compared with dbh-CB1-WT mice. (*n* = 6–9). * *p* ≤ 0.05. dbh-CB1-KO mice on HFD show increased *β3-adrenergic receptor* mRNA in BAT. For M, N, O, and P (*n* = 5–8), * *p* ≤ 0.05 by Student *t*-test. Data are expressed as mean ± SE.

**Figure 4 ijms-23-12635-f004:**
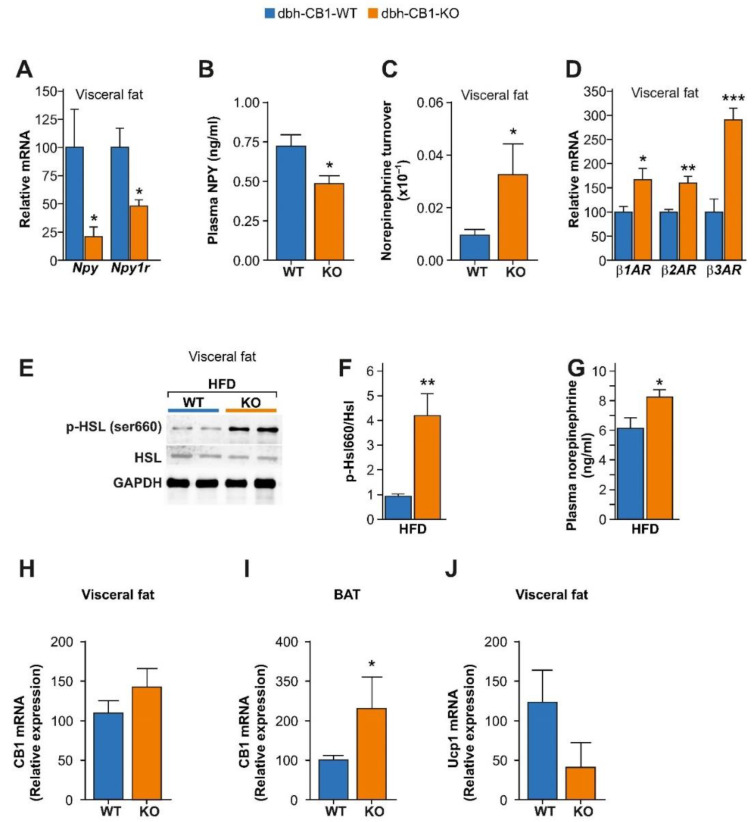
HFD selectively enhances β-adrenergic signaling while decreases NPY signaling in the visceral fat of dbh-CB1-KO mice. (**A**) qPCR analysis of visceral fat from dbh-CB1-KO mice on HFD show decreased *Npy* and *Npy1r* gene expression (*n* = 5–6). (**B**) Lower plasma NPY levels in dbh-CB1-KO mice on HFD (*n* = 7). (**C**) dbh-CB1-KO mice on HFD show increased norepinephrine turnover in the visceral fat (*n* = 6–8). (**D**) dbh-CB1-KO mice on HFD show increased *β1, β2, and β3 adrenergic* receptor mRNA levels in the visceral fat (*n* = 6). (**E**,**F**) Representative blot and quantification analysis showing the several-fold increase in p-Hsl/Hsl ratio in the visceral fat of dbh-CB1-KO mice on HFD (*n* = 3–4). (**G**) dbh-CB1-KO mice show increased plasma NE levels in HFD conditions (*n* = 5–7). *CB1* mRNA levels in (**H**) visceral fat and (**I**) BAT from dbh-CB1-KO and the corresponding control littermates. (**J**) *Ucp1* mRNA in visceral fat. Data are expressed in mean ± SEM. Student *t*-test, * *p* < 0.05, ** *p* ≤ 0.01, *** *p* ≤ 0.001.

**Figure 5 ijms-23-12635-f005:**
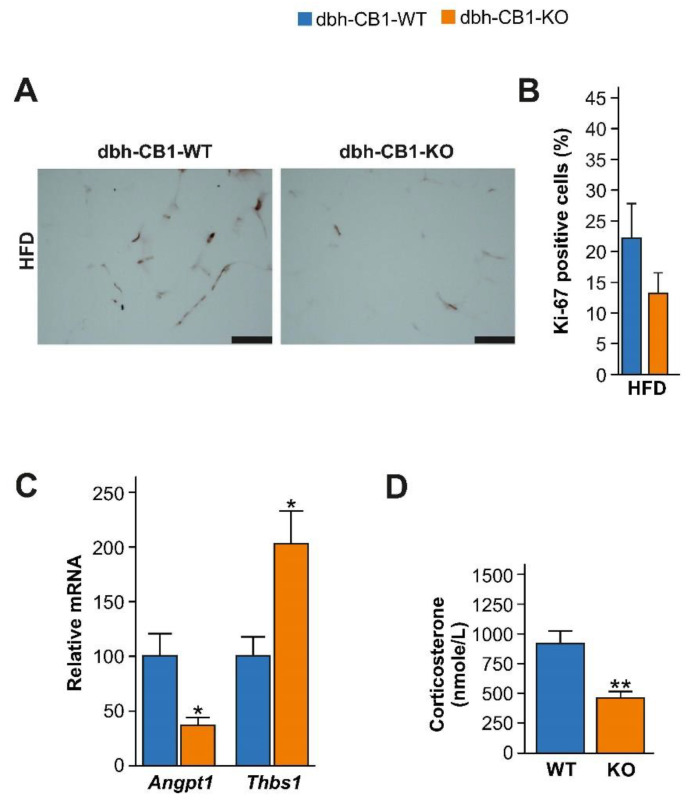
Reduced visceral adiposity in dbh-CB1-KO mice is associated with low angiogenesis and corticosterone levels. (**A**) Immunostaining against Ki 67, a proliferation marker, of visceral fat from mice on HFD. Scale bar = 100 μm. (**B**) Percentage of Ki-67 positive cells (*n* = 3), Student *t*-test. (**C**) qPCR analysis of *Angpt1* and *Thbs1,* markers of angiogenesis in visceral fat in HFD-fed mice (*n* = 5–7). (**D**) Plasma levels of corticosterone in mice on HFD (*n* = 5–7). ** *p* ≤ 0.01; * *p* ≤ 0.05 by Student’s *t*-test. Data are expressed as mean ± SE of the mice.

**Table 1 ijms-23-12635-t001:** TaqMan probes used for the real-time qPCR analysis.

Gene Name	Probe Code
Peroxisome proliferator-activated receptor-γ (*Pparg*)	Mm01184323_m1
CCAAT/enhancer-binding protein-α (*Cebpa*)	Mm00514283_s1
Leptin (*Lep*)	Mm00434759_m1
Fatty acid synthase (*Fas*)	Mm00662291_g1
Uncoupling protein-1 (*Ucp1*)	Mm01244860_m1
Cannabinoid receptor type 1 (*CB1*)	Mm00432621_s1
Glucoronidase-β (*Gusb*)	Mm00446956_m1
Cytochrome C oxidase subunit IV (*Cox4i2*)	Mm00446387_m1
Mitochondrial transcription factor (*Tfam*)	Mm00447485_m1
Hormone-sensitive lipase (*Lipe*)	Mm00495359_m1
Adrenergic receptor β_1_ (*β1AR*)	Mm00431701_s1
Adrenergic receptor β_2_ (*β2AR*)	Mm02524224_s1
Adrenergic receptor β_3_ (*β3AR*)	Mm02601819_g1
Dopamine β-hydroxylase (*Dbh*)	Mm00460472_m1
Neuropeptide Y (*Npy*)	Mm03048253_m1
Neuropeptide Y receptor 1 (*Npy1r*)	Mm00650798_g1
Angiopoetin 1 (*Angpt1*)	Mm00456503_m1
Thrombospondin-1 (*Thbs1*)	Mm00449032_g1
PR domain containing 16 (*Prdm16*)	Mm00712556_m1
Citrate synthase (*Cs*)	Mm00466043_m1
CCAAT/enhancer-binding protein-β (*Cebpb*)	Mm00843434_s1
TATA binding protein (*Tbp*)	Mm00446973_m1

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
