# Peer review of "Cannabinoid CB1 Receptor Deletion from Catecholaminergic Neurons Protects from Diet-Induced Obesity"

_ijms, 2022, doi:10.3390/ijms232012635_

Round 1

Reviewer 1 Report

The authors tested the hypothesis that CB1 receptors (CB1) located on DBH-expressing sympathetic neurons regulate SNS activity and are required for diet-induced obesity (DIO) in mice. The hypotheses was tested by analyzing DIO and related metabolic and hormonal changes in mice with conditional knockout of CB1in dbh-expressing neurons and their wild-type littermates.

Comments: The data in this manuscript provide convincing evidence for a key role of CB1 on sympathetic postganglionic nerve terminals in DIO and its metabolic/hormonal sequelae. The authors should clarify and discuss the following issues.

1. In an earlier report (Ruiz de Azua, ref. 25) the authors presented convincing evidence indicating that conditional KO of CB1 in adipocytes results in resistance to DIO, very similar to that shown here in dbh-CB1-KO mice. These two sets of findings need to be reconciled. For example, it has been shown that induction of BAT thermogenesis by beta-3 adrenoceptor activation or cold exposure reslts in upergulation of CB1expression and EC production in BAT (J Lipid Res 57:464-73, 2016; Diabetes 67:1226-36, 2018), which represent a negative feedback mechanism that limits the increase in energy expenditure. Eliminating this obesity-promoting mechanim by conditional CB1 KO in WAT and BAT would make mice more resistant to DIO. The authors should check if CB1 expression is increased in BAT and WAT of dbh-CB1-KO compared to wt mice, which would be compatible with this explanation. A further likely consequence of increased SNS activity in dbh-CB1-KO mice is browning or beige-ing  of WAT, which could also be easily tested. The suggested experiments are easy to do using existing tissue samples. This issue should definitely be addressed and discussed.

2. NPY is a co-transmitter of NE in sympathetic neurons, and although they are not always released in proportional amounts, the robustly opposite changes of plasma NPY and NE reported here are counterintuitive. Interestingly, in ref. 37 the authors reported an increase in NPY levels in bone of dbh-CB1-KO vs control mice, which is opposite to its reduced levels in plasma and visceral fat (see Fig 4A,B). Some explanation is needed.

3. It is assumed but not stated anywhere in the manuscript whether controls were Cre-negative littermates (rather than unrelated wild-type mice). This should be stated in the Methods section.

4. At the end of the Discussion, the authors correctly state that their findings 'suggest a potential role of peripheral CB1 in obesity and obesity-related disorders. They should add that these findings are compatible with the well documented anti-DIO efficacy of non brain-penetrant CB1 antagonists and their therapeutic potential in visceral obesity.     

Author Response

Reviewer 1

Comment: In an earlier report (Ruiz de Azua, ref. 25) the authors presented convincing evidence indicating that conditional KO of CB1 in adipocytes results in resistance to DIO, very similar to that shown here in dbh-CB1-KO mice. These two sets of findings need to be reconciled. For example, it has been shown that induction of BAT thermogenesis by beta-3 adrenoceptor activation or cold exposure results in upergulation of CB1 expression and EC production in BAT (J Lipid Res 57:464-73, 2016; Diabetes 67:1226-36, 2018), which represent a negative feedback mechanism that limits the increase in energy expenditure. Eliminating this obesity-promoting mechanism by conditional CB1 KO in WAT and BAT would make mice more resistant to DIO.

The authors should check if CB1 expression is increased in BAT and WAT of dbh-CB1-KO compared to wt mice, which would be compatible with this explanation. A further likely consequence of increased SNS activity in dbh-CB1-KO mice is browning or beigeing of WAT, which could also be easily tested. The suggested experiments are easy to do using existing tissue samples. This issue should definitely be addressed and discussed

Reply: We are thankful to the reviewer for this suggestion. Accordingly, we checked CB1 mRNA levels in BAT and WAT of dbh-CB1-KO mice compared to WT mice and found a non-significant increase in EWAT but a significant increase in BAT of mutant mice as compared to control mice. We also checked Ucp1 mRNA levels to examine the beigeing of WAT. However, we could not detect significant differences. These data are shown in Figure 4 H-J.

Comments: NPY is a co-transmitter of NE in sympathetic neurons, and although they are not always released in proportional amounts, the robustly opposite changes of plasma NPY and NE reported here are counterintuitive. Interestingly, in ref. 37 the authors reported an increase in NPY levels in the bone of dbh-CB1-KO vs control mice, which is opposite to its reduced levels in plasma and visceral fat (see Fig 4A,B). Some explanation is needed.

Reply: We have previously reported increased NPY levels in the bone of aged dbh-CB1-KO mice on a standard diet as compared to wild-type controls. However, in the current study, we found that high-caloric diet treatment enhanced the release of NE instead of NPY in the mutant mice as compared to control mice. Based on our findings, we propose that CB1 deficiency enhances the activity of dbh+ neurons thereby increasing the co-release of NE and NPY from sympathetic terminals, but the NE/NPY ratio can change under certain circumstances (here: HFD, or acute vs chronic stress, aging, etc.). For example, in Kuo et al. (2007), the authors reported that stress led to the increased release of NPY from the sympathetic nerves.

Comments: It is assumed but not stated anywhere in the manuscript whether controls were Cre-negative littermates (rather than unrelated wild-type mice). This should be stated in the Methods section.

Reply: More details about the control mice were added in material and methods section.

Comments: At the end of the Discussion, the authors correctly state that their findings suggest a potential role of peripheral CB1 in obesity and obesity-related disorders. They should add that these findings are compatible with the well-documented anti-DIO efficacy of non-brain-penetrant CB1 antagonists and their therapeutic potential in visceral obesity

Reply: Relevant references and findings were added in the revised version.

Reviewer 2 Report

Manuscript Number: ijms-1839940

This manuscript describes the characterization of metabolic phenotypes of a mouse model that lacks the cannabinoid type 1 receptor (CB1) in dopamine β-hydroxylase (dbh)-expressing cells including sympathetic ganglia, adrenal medulla, and catecholaminergic neurons. The authors found that the mutant mice were protected from diet-induced obesity due to the increased energy expenditure. Also, the mice showed increased sympathetic nervous system (SNS) tone and decreased basal HPA axis activity. They concluded that the deletion of CB1 from dbh cells increases SNS activity, and protects mice against diet-induced obesity.

In my view, the subject of the manuscript is timely and interesting, and the results are presented in a form that is overall appropriate and sound. In particular, the novelty of this work lies in identifying the role of CB1 in dbh cells, although the studies in this manuscript are largely descriptive. My main concern with the paper is that the metabolic phenotype may arise from the altered functions of the adrenal gland rather than from the altered CB1 functions in the catecholaminergic neurons, which is the authors’ conclusion. To further bolster the authors’ claim, the following would be suggested: examine if CB1 is deleted in adrenal grand; determine if altered CB1 signaling in the adrenal gland modulates the adrenal structures/functions. This can be achieved by examining the gross morphology of the adrenal gland and also by assessing basal and stress-induced corticosterone levels. As Dbx-Cre is expressed not only by neurons but also by the periphery, the authors may want to be careful in concluding that the effect is due to the deletion of CB1 in catecholaminergic neurons, unless the authors could exclude the possibility that the adrenal gland may play a role.

Are there sex differences in the phenotypes? 

Fig. 3: There is general agreement that data on energy expenditure should not be corrected for body weight. The data should be presented independent of body mass or should be ANCOVA corrected for body mass such as Fig.3 H.

Author Response

Reviewer 2

Comment: My main concern with the paper is that the metabolic phenotype may arise from the altered functions of the adrenal gland rather than from the altered CB1 functions in the catecholaminergic neurons, which is the authors’ conclusion. To further bolster the authors’ claim, the following would be suggested: examine if CB1 is deleted in the adrenal gland; determine if altered CB1 signaling in the adrenal gland modulates the adrenal structures/functions. This can be achieved by examining the gross morphology of the adrenal gland and also by assessing basal and stress-induced corticosterone levels. As Dbx-Cre is expressed not only by neurons but also by the periphery, the authors may want to be careful in concluding that the effect is due to the deletion of CB1 in catecholaminergic neurons, unless the authors could exclude the possibility that the adrenal gland may play a role.

Reply: In the result section, we stated that CB1 mRNA levels were reduced (50%) in the whole adrenal gland in KO mice and that a larger reduction (about 75%) was found in the isolated adrenal medulla. Furthermore, we found a decrease in plasma CORT levels in HFD-fed KO vs WT mice (Fig 4D). Therefore, changes in the adrenal function might play an important role in weight regulation in these mice as stated in the abstract and the main text, although, at this time, we cannot exclude that increased SNS tone concomitant with reduced NPY activity can also contribute to the metabolic phenotype of dbh-CB1-KO mice. We proposed that CB1 deficiency in dbh-positive cells protects from HFD-induced weight gain via multiple mechanisms.

Comment: Are there sex differences in the phenotypes? 

Reply: We did not analyze the metabolic parameters in females.

Comment: Fig. 3: There is general agreement that data on energy expenditure should not be corrected for body weight. The data should be presented independent of body mass or should be ANCOVA corrected for body mass such as Fig.3 H.

Reply: The data on energy expenditure experiments are now presented without correction for body weight.